# Advances in the Integrated Pest Management of Quinoa (*Chenopodium quinoa* Willd.): A Global Perspective

**DOI:** 10.3390/insects15070540

**Published:** 2024-07-18

**Authors:** Luis Cruces, Eduardo de la Peña, Patrick De Clercq

**Affiliations:** 1Department of Entomology, Faculty of Agronomy, Universidad Nacional Agraria La Molina, Lima 12-056, Peru; 2Department of Plants and Crops, Faculty of Bioscience Engineering, Ghent University, 9000 Ghent, Belgium; e.delapena@csic.es (E.d.l.P.); patrick.declercq@ugent.be (P.D.C.); 3Instituto de Hortofruticultura Subtropical y Mediterránea “La Mayora” (IHSM-UMA-CSIC), Spanish National Research Council (CSIC), Estación Experimental “La Mayora”, Algarrobo-Costa, 29750 Malaga, Spain

**Keywords:** quinoa, pests, IPM, pesticides, biological control

## Abstract

**Simple Summary:**

Quinoa is an Andean grain that has gained global popularity due to its nutritional properties. The production area has expanded outside its Andean origins and the crop is now produced worldwide. However, new emerging pests cause serious problems in the crop and limit its production. This review discusses the status of the pests of quinoa in the world and the agricultural strategies implemented in the framework of integrated pest management (IPM). The literature regarding sampling methodologies for pest monitoring, determining economic threshold levels, cultural practices, and chemical and biological control is summarized, and a critical perspective on establishing IPM in quinoa is given.

**Abstract:**

Since ancestral times, quinoa (*Chenopodium quinoa* Willd.) has been cultivated in the Andean regions. Recently, this pseudocereal has received increasing international attention due to its beneficial properties, such as adaptation and resilience in the context of global change, and the nutritional value of the grains. As a result, its production areas have not only increased in the highlands of South America but have also expanded outside of its Andean origins, and the crop is currently produced worldwide. The key pests of quinoa in the Andean region are the gelechiid moths *Eurysacca melanocampta* and *Eurysacca quinoae*; in other parts of the world, new pest problems have recently been identified limiting quinoa production, including the gelechiid *Scrobipalpa atripicella* in North America and Europe and the agromyzid fly *Amauromyza karli* in North America. In this review, the status of quinoa pests in the world is presented, and different aspects of their integrated management are discussed, including sampling methodologies for pest monitoring, economic threshold levels, and various control strategies.

## 1. Introduction

Quinoa, *Chenopodium quinoa* Willd. (Amaranthaceae), is a pseudocereal native to the Andean region, an extensive mountain chain that runs from the north to the south of South America, through the countries of Venezuela, Colombia, Ecuador, Peru, Bolivia, Chile, and Argentina, with an average altitude between 3000 and 4000 m a.s.l. [1,2,3,4].

For a long time, quinoa was an underutilized species, even considered by the local population as a marginal crop, and its consumption was associated with the poorest segments of society [5,6,7]. However, as a result of the increasing attention from the international scientific community, the multiple benefits of this crop have been unraveled (e.g., tolerance to drought, salinity, and cold; high nutritional value; and adaptation to different environments) [6,7,8]. Additionally, quinoa has received considerable interest from the international market due to its nutritional properties (i.e., rich in lysine, vitamins, calcium, magnesium, copper, phosphorous, potassium, and zinc, and an absence of gluten), and in particular, has drawn attention from countries where nutritional problems are endemic issues [8,9,10]. In this context, the demand for quinoa has increased significantly in recent years, leading to an extraordinary expansion of the cultivated area in the South American countries of Ecuador, Chile, and particularly Bolivia and Peru (the main quinoa producers) and also worldwide outside of its origins [11].

When a new crop is introduced, the occurrence of pests previously unknown for the zone may become a limiting factor for crop success. Hence, insights gained from pest management practices in regions facing similar pest challenges can offer valuable lessons for developing strategies for integrated pest management (IPM). Implementing pest monitoring systems and adequate cultural practices and exploring better alternatives to chemical and biological control in the frame of an IPM program may support the long-term sustainability of quinoa production [12]. This implies a suitable combination of pest control methods to reduce economic crop damage by preventing and suppressing pests in a cost-effective manner and with the least possible environmental hazard [13,14].

In this review, we compile the relevant literature on pests that currently threaten quinoa production around the world and revise the management strategies that have been used against these pests from an IPM perspective.

## 2. Global Expansion of Quinoa

Quinoa is a crop that has been cultivated since ancient times in the Andean region, mainly in Bolivia and Peru and to a lesser extent in Argentina, Chile, and Ecuador. This pseudo-cereal was one of the main foods for the Andean settlers in the pre-Hispanic period until the South American conquest period, when the introduction of cereals, such as wheat, barley, and oats, surrogated quinoa. As a result, the cultivated area of quinoa suffered a dramatic reduction, particularly in the inter-Andean valleys of the central and northern highlands (2300–3500 m a.s.l.). However, its production at higher elevations (over 3500 m a.s.l.), has been maintained to present; consequently, this agroecological zone has the highest genetic diversity of quinoa [8,15,16,17,18,19].

Throughout most of this period, quinoa was considered a marginal crop, primarily used in subsistence farming, for farmer family consumption, or as fodder for animals (i.e., cattle, pork, sheep, and chicken). It was mainly cultivated by smallholder farmers in plots usually less than two hectares in size. This situation still persists in a large part of the Andes [15,16,18]. However, over the last two decades, quinoa has gained international interest due to its resilience and high nutritional value [10,20,21,22]. By the end of the 1980s, the crop was produced in 11 countries outside of the Andes. By the end of 2012, 30 more countries cultivated this Andean grain for both research and production. In 2013, the Food and Agriculture Organization of the United Nations (FAO) declared the International Year of Quinoa (IYQ), and significant efforts were made during this year to promote quinoa consumption [6,11,23,24]. Subsequently, by 2018, the crop was cultivated for research and production in 123 countries [11].

With higher quinoa demands, the national and international market value increased, and export prices rose more than fivefold, making quinoa a more profitable crop. This motivated more farmers in South America to grow this grain, even at lower elevations, leading to a considerable increase in the cultivation area. In Peru, the quinoa harvested area increased from 28,889 ha (2000) to 69,202 ha (2022); in Bolivia, it increased from 36,847 ha (2000) to 123,627 ha (2022) [24,25].

Nowadays, food trends and diet shifts in the global north have created market windows where organic quinoa, which has a higher price in the market as compared with conventional quinoa, is considered an important source of high-quality plant-based proteins [11].

## 3. Quinoa Cropping Systems

### 3.1. Traditional System

Quinoa is cultivated under a traditional system in the Altiplano or inter-Andean valleys in South America (2500–4000 m a.s.l.), where production is mostly organic. In this system, there is an intensive use of labor from soil preparation (sometimes aided by animal traction) to harvest (Figure 1). This cultivation is mostly rainfed, although there are accessible zones near the rivers where complementary surface irrigation can be applied [19].

Traditional quinoa cultivation usually involves a crop rotation system or crop associations with corn, beans, broad beans, cucurbits, potatoes, tarwi, and other native grains and tubers (Figure 1A). A common agricultural practice is to prepare the soil just after the harvest in order to take advantage of the humidity that remains from the rainy season. After preparation, farmers leave the field for 2 to 4 months until the beginning of the next cropping season. Another regular practice is organic fertilization, with the incorporation of manure from the cattle in the fields [11,19,22].

### 3.2. Modern System

Modern systems involve monocultures that can be grown under organic (mainly in the Andean region) or conventional production systems. Recently, more advanced farming techniques have been applied in quinoa production, in zones outside of the highlands of South America (at lower elevations) and in other parts of the world. Adapted agricultural machinery is applied in soil preparation, seeding, cultivation, hilling, weeding, and harvest; irrigation can be surface or technified, and the use of chemical inputs (fertilizers and pesticides) is high and systematic, which may include modern equipment (Figure 2) [19,24,26]. In some cases, farmers use transplanting to establish the crop, in order to reduce weed problems [27,28].

## 4. Major Pests of Quinoa

Around 80 arthropods have been recorded to infest quinoa in South America, but most of these phytophagous species are of minor importance. In other parts of the world where quinoa is cultivated for research and/or production, the pest complex differs from that in its Andean origins; however, there are still few studies on the pests that infest quinoa plants, and most of the reports refer only to their presence in the crop, which is probably due to the fact that the damage caused is considered minor. In Appendix A), we present a list of phytophagous arthropods that have been reported to occur so far on quinoa worldwide [24,29,30,31,32].

### 4.1. Piercing-Sucking Insects

Piercing-sucking insects that reportedly infest quinoa belong to the groups of aphids, rhopalids, mirids, lygaeids, and thrips. In South America, the potato aphid, *Macrosiphum euphorbiae* (Thomas) (Hemiptera: Aphididae) infests the underside of the leaves, flower primordia, and inflorescences. The true bugs *Liorhyssus hyalinus* (Fabricius) (Hemiptera: Rhopalidae) and *Nysius simulans* Stål (Hemiptera: Lygaeidae), feed on different plants parts, but usually the highest infestation occurs during the grain formation. The western flower thrips, *Frankliniella occidentalis* (Pergande) (Thysanoptera: Thripidae), has been observed from the flowering stage to ripening in high densities, although injury in terms of yield reduction has not been reported [24,33,34,35,36,37].

In North America, various plant bugs (Hemiptera: Miridae) have been reported from quinoa in different states of the USA, including *Melanotrichus coagulatus* (Uhler), which has been noted attacking seedlings, and *Lygus hesperus* Knight and *Lygus elisus* Van Duzee reportedly causing seed abortion and consequent yield losses [28,38]. The lygaeid *Nysius raphanus* Howard has been noted attacking seedlings. Large populations of *Hayhurstia atriplicis* (L.) (Hemiptera: Aphididae) have been reported to cause damage on the developing foliage; another aphid, *Pemphigus populivenae* Fitch was observed on quinoa roots, leading to yield declines [28,38,39]

In Southern Europe, high infestation of *Nysius cimoydes* (Spinola) (Hemiptera: Lygaeidae) on quinoa panicles was observed in Italy, feeding on grains close to the harvest. *Lygus rugulipennis* (Poppius) and *Orthotylus flavosparsus* (Sahlberg) (Hemiptera: Miridae) were reported as the most abundant pests in experimental fields [40,41].

In Morocco, the most recurrent species were observed to be the black bean aphid *Aphis fabae* Scopoli and the green peach aphid *Myzus persicae* (Sulzer) (Hemiptera: Aphididae), as well as the southern green stinkbug *Nezara viridula* (L.) (Hemiptera: Pentatomidae) [42,43].

### 4.2. Stem Borers

*Cosmobaris americana* (Coleoptera: Curculionidae) has been reported causing considerable damage on the stem and petioles in the USA [28]. In 2022, the stem boring fly *Amauromyza karli* Hendel (Diptera: Agromyzidae) infested up to 100% of crops, causing significant damage [44].

### 4.3. Defoliators

In South America, *Spodoptera eridania* (Cramer) (Lepidoptera: Noctuidae) has been reported to feed on the foliage and developing grains [24,45], whereas in North America, *Spodoptera exigua* (Hübner) was observed to cause extensive defoliation [38].

In Denmark, serious infestations of *Cassida nebulosa* L. (Coleoptera: Chrysomelidae) were observed, causing damage to the leaves in experimental plots of quinoa [46]. High infestation by *Chaetocnema tibialis* (Illiger) (Coleoptera: Chrysomelidae) was reported in experimental fields in Italy [41].

### 4.4. Grain Borers

In the Andean region, *Eurysacca melanocampta* (Meyrick) and *Eurysacca quinoae* Povolný (Lepidoptera: Gelechiidae) are the most important pests of quinoa. The caterpillars attack the plants at the early stages, produce leaf mines, and spin the apical leaves together and feed on the apical buds inside. However, the main damage is produced at later crop stages, when the larvae infest the panicles to feed on the developing grains. Other lepidopteran pests that can be of major importance in the Andean regions are *Helicoverpa quinoa* Pogue and Harp, *Copitarsia incommoda* (Walker), *Chloridea virescens* (Fabricius) (Lepidoptera: Noctuidae), and *Spoladea recurvalis* (Fabricius) (Lepidoptera: Crambidae), all of them feeding on the developing grains inside the panicle [24,29,30,31,32,34,47,48].

*Scrobipalpa atriplicella* (Fischer von Röslerstamm) (Lepidoptera: Gelechiidae), which produces similar damage to *E. melanocampta*, has been reported to cause serious problems in the USA and Canada, as well as in Denmark [46,49,50]. It has been documented that two generations of this pest per year may occur on quinoa in Europe: the first in spring (May–June) and the second in summer (July–August). The caterpillars feed cryptically on the seed, inside the panicle [49].

### 4.5. Leaf Miner Flies

High infestation of *Liriomyza huidobrensis* (Blanchard) (Diptera: Agromyzidae) was observed in South America, causing economic damage. The larvae feed in the spongy mesophyll tissue and produce long twisting mines in the leaves, leading to leaf discoloration and senescence, and eventually defoliation [34,37].

## 5. Integrated Pest Management in Quinoa

Integrated pest management (IPM) refers to coordinated strategies of pest management to keep pest densities at levels below economic thresholds. An IPM approach combines methods of pest control, taking advantage of natural mortality, and considering the least possible disruption to agroecosystems and as minimal risks as possible to humans and the environment [51].

IPM is based on eight principles: (i) prevention and suppression of pests; (ii) monitoring; (iii) decision-making based on monitoring and thresholds; (iv) combining control measures, keeping chemical methods as the last option; (v) pesticide selection; (vi) reduced pesticide use; (vii) anti-resistance strategies; and (viii) evaluation [52]. Based on these principles, the current strategies applied in the integrated production of quinoa are summarized.

The information presented below comprises strategies of pest control mostly applied in Bolivia and Peru, which can serve as a reference when implementing IPM programs outside of the Andean region.

### 5.1. Monitoring

#### 5.1.1. Plant Sampling

In the Andean region, quinoa farmers are generally recommend to sample 10–25 plants per ha. Monitoring should be systematic throughout the crop phenology, but particular attention should be paid from flowering to grain maturation. Quinoa plants from the borders should always be avoided for sampling, because they are near roads, ditches and nearby crops, and therefore they are not representative of the situation within the field [24,32].

The sampling units to take into consideration may include the following [24,44]:Whole plant (at germination), to monitor initial damage of agromyzid adults.The stem, during the vegetative stage to detect the first generation of stem boring flies (e.g., A. karli).Leaves (during the vegetative stage), to monitor leafminer flies, aphids, and lepidopteran larvae.Flower primordia and leaves (at the beginning of flowering), to monitor lepidopteran larvae and leafminer flies.Whole panicle (from flowering to maturation), to monitor lepidopteran larvae, plant bugs, lygaeids, rhopalids, and thrips.

#### 5.1.2. Use of Traps

The use of colored sticky traps, molasses traps and pheromone traps has been incorporated in the management of quinoa pests, albeit not extensively [24,32,49].

Color and molasses traps

Colored sticky traps can be placed at the borders of the field, aimed mainly to monitor adults of agromyzids (i.e., *L. huidobrensis, A. karli*), thrips, aphids, and heteropterans. Due to the structure of the quinoa plants and the high plant density used, mass trapping cannot be implemented [34,44].

Molasses traps have also been included to monitor pests, particularly adults of Lepidoptera. In the Andean region, such traps are handcrafted by recycling bottles and containers of different sizes, using molasses and water in a ratio of 1:1. Lepidopteran pests, such as noctuids, pyralids, and gelechiids, are attracted by the sweet odor of the molasses, and they are captured by its sticky consistency. However, other non-pest lepidopterans can also be attracted, and the evaluator may confuse them with the quinoa pests; in particular, the smallest moths may be difficult to distinguish from *E. melanocampta* or *S. atripicella*, which easily lose their wing scales when trapped. Therefore, this kind of trap is more useful to monitor larger-sized moths, such as noctuids and some pyralids [24].

b.Pheromone traps

In Bolivia, the use of pheromone traps has been reported for noctuid moths, with commercially available synthetic pheromones for *H. quinoa, C. incommode*, and *Agrotis andina* Köhler. Four pheromone traps per hectare are recommended to monitor the presence of the adults of each of these species. In zones where population densities of these noctuids are low, the use of four pheromone traps per hectare is allowed to keep larval damage below economic levels [32,49].

The use of pheromone traps in the lowlands of Peru to monitor noctuids, such as *C. virescens* or *S. eridania*, has not been incorporated in the management of quinoa pests, although synthetic pheromones for *C. virescens* have recently been made commercially available in the country [53].

In North America, pheromones for *S. atripicella* have also been developed, although they are not yet commercially available to date [49].

### 5.2. Economic Thresholds

Economic thresholds have been defined for a very limited number of quinoa pests. In Bolivia, when numbers of *H. quinoa* larvae average more than one individual per plant, it is recommended to apply control measures [32], whereas an economic threshold level of three–six larvae per plant is suggested for *E. melanocampta* or *E. quinoae* [54]. Economic thresholds for quinoa production in other parts of the world have not been reported.

### 5.3. Cultural Measures

#### 5.3.1. Soil Preparation

Either through traditional practices (in the Andes) or by using machinery, soil preparation contributes to reducing cutworms and other pests that pupate in the soil (i.e., some lepidopterans and dipterans). In the coastal areas of Peru, farmers usually perform irrigation before soil preparation to facilitate the use of tillage tools. This practice also contributes to eliminating remaining pest stages from the previous crop and promotes the germination of weeds that later are destroyed during the tillage or mechanical weeding [8,24,32].

#### 5.3.2. Plant Density

In the Andean region, quinoa is usually manually sown aided with perforated small containers that allow the seed to pass through, using more seeds than necessary (10–20 kg/ha); this results in a high density of germinated plants. This occurs mainly due to the limited access to sowing machines in the traditional production areas, which significantly reduces the number of seeds used when sowing (4–5 kg/ha). However, this high stand density of quinoa allows the farmers to overcome initial damage caused by cutworms, and about two weeks after germination they eliminate the excess of plants in order to obtain a suitable plant density, depending on the variety used [8,24].

#### 5.3.3. Sowing Periods

Under Andean conditions, quinoa is rainfed, therefore the sowing period takes place when the rainy season begins. This allows it to have a long fallow period during the dry months, when farmers cannot cultivate any crops. In this way, the life cycle of insect pests can be interrupted, and soil fertility is not depleted [19,32].

In temperate areas, quinoa is cultivated during a warm period, from about spring to summer, and during the coldest period of the year most pests are in a dormant stage [6,28,55].

#### 5.3.4. Cultivation and Hilling

Either manually (in the Andes) or by using machinery, cultivation and hilling are common practices in quinoa, aimed at improving aeration in the soil and providing better support to the base of plants. With these practices, larvae and pupae of insect pests, alongside weeds, are suppressed [6].

#### 5.3.5. Weed Control

Currently, there are no specific herbicides registered for use in quinoa and therefore, weed control relies on mechanical and manual methods [8,55].

Manual weeding is performed during the critical period, 2–3 weeks after germination, to minimize weed competition and during the flowering stage [8,24,55]. In areas and periods of the year where competitive weeds rapidly emerge, the use of a seedbed established 3–4 weeks before plantation can be an option to reduce problems with weeds during the first critical period of the crop [55]. Weeds are also controlled during the tilling, cultivation and hilling practices [24]. Inter-row hoeing of row-spaced quinoa (0.5 m) can be performed at the beginning of the crop phenology [55].

Systemic herbicides, like glyphosate, may be applied, but only in the field borders and near irrigation ditches [8].

#### 5.3.6. Crop Rotation and Intercropping

Quinoa is a potential crop to be used in crop rotations because it uses less nitrogen than cereals like wheat and barley [28]. However, currently there is no specific information on crop rotation systems applied in quinoa production worldwide.

In the Andean region, farmers usually consider other native crops that have long been part of their traditional staple crops. These include tubers (e.g., potato, oca, and maca), grains, corn, and a range of legumes (e.g., broad bean and tarwi). This practice allows to interrupt the life cycle of insect pests and increases the benefits of biodiversity in the agroecosystem [19,32,56,57,58]. In the traditional cultivation system, intercropping with these native crops is also incorporated as a cultural practice by farmers from the highlands [19].

### 5.4. Use of Synthetic Chemical Insecticides

In conventional quinoa production, farmers from the Andean countries usually rely on broad-spectrum insecticides such as pyrethroids (e.g., cypermethrin, alphacypermethrin, and lambda-cyhalothrin), organophosphates (e.g., chlorpyrifos, methamidophos, dimethoate, and monocrotophos) and carbamates (e.g., carbaryl, cartap, and methomyl) [8,26,32,34]. These pesticides are mostly used without a rotation system, and sometimes they are applied following a fixed schedule of treatments (i.e., calendar spraying) rather than a system based on the actual infestation level; this is performed up to 60 days after germination in order to reduce the risks of harvests being contaminated with chemical residues [26,34].

The use of organophosphates has been increasingly phased out due to risks to human health and the environment [59]. Additionally, given the broad spectrum of pyrethroids and carbamates, which kill not only pests but also their natural enemies, their usability in an IPM context is limited [59,60].

The use of neonicotinoids such as imidacloprid or acetamiprid has been noted to reduce the incidence and damage of leafminer flies such as *L. huidobrensis* and *A. karli* [44]. However, imidacloprid and other neonicotinoids have a negative environmental profile, and their side effects on natural enemies and bees have been widely documented [61,62,63,64]. Soil applications of acetamiprid at the early stages of the crop, when fewer natural enemies are established in the field, may assist in slowing down the population buildup of sap-feeding insects like aphids, rhopalid, or lygaeids [44].

There are a wide variety of more selective insecticides with less harmful effects on humans and non-target organisms in the environment available on the market. Compounds like emamectin benzoate, lufenuron, teflubenzuron, and chlorantraniliprole may be part of pest management to control lepidopteran larvae, under a rotation system based on their mode of action [8,32,53,61,65].

The surface of the leaves, stem, and panicle of quinoa plants is heavily covered with calcium oxalates, which reduce the adherence of the foliar applications. Therefore, it is recommended to use adjuvants to improve pesticide coverage [66].

### 5.5. Use of Bioinsecticides

Environmentally friendly insecticides include microbial formulations, bacterial fermentation products, and botanicals. In Peru, *Bacillus thuringiensis kurstaki, B. thuringiensis aizawai*, and spinosad are sprayed against certain species of lepidopteran larvae, particularly in organic quinoa production. Tetraethyl silicate has been used against aphids (Appendix A). To enhance the adherence of these bioinsecticides to the foliage, the use of adjuvants is recommended [8,32,53].

### 5.6. Biological Control

#### 5.6.1. Natural Biological Control

In its area of origin, quinoa is visited by a wide variety of predators and parasitoids that function as natural biological control agents of herbivores [29,34,67]. The reported predator and parasitoid species whose host range includes pests of quinoa in the Andean region is compiled and presented in the Appendix A) and may serve as a reference for other parts of the world [24,32,34,67,68].

Predators

The most common predators found in quinoa in the Andean region belong to the families Carabidae, Coccinellidae, Staphylinidae, Labiduridae, Syrphidae, Dolichopodidae, Anthocoridae, Berytidae, Geocoridae, Nabidae, Pentatomidae, Chrysopidae, and Hemerobiidae and spiders of the families Thomisidae, Oxyopidae, Lycosidae, Dysderidae, and Salticidae [24,67]. There is no information on the role of predatory mites in quinoa fields.

Larvae and adults of ground beetles (Carabidae) feed on epigeous insects such as cutworms, crickets, and caterpillars of noctuids that go down from the foliage to the soil to pupate [69]. Adults and larvae of ladybirds (Coccinellidae) are avid predators of aphids and other small, soft-bodied insects. The predatory rove beetles (Staphylinidae) and earwigs (Labiduridae, Forficulidae) mainly prey on soil surface-dwelling arthropods. Predatory species of hoverflies (Syrphidae) prey in the larval stage on aphids, whereas adults feed on nectar and pollen. Adults of long-legged flies (Dolichopodidae) feed on small insects and mites; they have been observed preying on agromyzid adults such as *L. huidobrensis*. Anthocorids, berytids, geocorids, and nabids also feed on small insects, mainly aphids and thrips, as well as on eggs or the first instars of lepidopterans and on mites. Predatory pentatomids usually feed on lepidopteran and some coleopteran larvae. Larvae of green lacewings (Chrysopidae) and larvae and adults of brown lacewings (Hemerobiidae) are important predators of aphids, whiteflies, thrips, eggs, and small larvae of lepidopterans [24,32,34,67].

Among the insect predators, species of the family Coccinellidae and Chrysopidae are noted to have a more significant role in pest regulation in Andean quinoa fields [67]. *Chrysoperla externa* (Hagen) has been reported to reach its highest population densities, coinciding with the occurrence of *Eurysacca* species [67,70]. Among the coccinellid species, *Hippodamia convergens* Guérin-Méneville and *Eriopis connexa* (Germar) have been reported during peak numbers of aphids [71]. Large numbers of larvae of the syrphid *Allograpta exotica* (Wiedemann) have been noted during high infestations of *M. euphorbiae* in Lima, Peru [34].

b.Parasitoids

The most common dipteran parasitoids belong to the family Tachinidae, which parasitize lepidopteran larvae and, to a lesser degree, heteropterans. Hymenopteran parasitoids reported from quinoa belong to the families Braconidae, Ichneumonidae, Scelionidae, Pteromalidae, Eulophidae, and Trichogrammatidae [24,32,34].

In the Andean region, parasitism of *E. melanocampta* in quinoa may reach up to 25% during the beginning of the grain filling, up to 45% during the milk grain stage, and up to 80% during the maturation of the grains [72]. The main larval parasitoids of *Eurysacca* moths are reported to be an unidentified species of the genus *Phytomyptera* (Tachinidae) and another of the genus *Diadegma* (Ichneumonidae), as well as *Copidosoma gelechiae* Howard (Encyrtidae) [31,32]. Parasitism observed for *L. huidobrensis* leafminers may reach up to 100%, mainly by *Halticoptera* spp. (Pteromalidae) and *Chrysocharis* spp. (Eulophidae), whereas for *M. euphorbiae*, only 2.5% of parasitism was observed, and the main collected parasitoids were *Lysiphlebus testaceipes* (Cresson), *Aphidius matricariae* (Dalman) and *Aphidius colemani* (Dalman) (Braconidae) [34].

#### 5.6.2. Applied Biological Control

The current published information suggests the potential for incorporating biological control strategies into an integrated pest management (IPM) program for quinoa.

Given the diversity of beneficial insects observed in quinoa fields [68], conservation biological control (CBC) programs can be feasible to help prevent pest outbreaks. This involves a series of practices that enhance the preservation of the established natural enemies in the crop and promote their population growth. Thus, providing a better environment for the natural enemy complex in the agroecosystem is expected to contribute to pest population regulation by the predators or parasitoids [73,74].

Augmentative Biological Control (ABC), in which mass-reared natural enemies are released in the crop for temporary pest suppression, is a strategy that should be explored for its potential for pest management in quinoa [75,76,77]. Chrysopid predators may be good candidates for the biological control of grain-feeding insects when no chemical insecticides can be applied [67,78]. For example, *C. externa* killed significant numbers of early stages of *L. hyalinus* and *N. simulans* in the laboratory and might thus help in protecting the crop at the grain filling stage. Additionally, releases of lacewings may assist in regulating populations of other species such as aphids, thrips, and small lepidopteran larvae that infest the quinoa fields [78,79,80,81].

Treatments with the entomopathogenic fungus *Beauveria bassiana* have also been recommended to control *C. virescens* and *F. occidentalis*, particularly at the beginning of the infestation [8,53]. Formulations based on granulosis and nucleopolyhedrosis viruses have been tested against *E. melanocampta*, with up to 50% control [67].

## 6. Discussion

While quinoa cultivation has expanded worldwide over the last decades, new pest problems have arisen, causing damage to grain production. In this context, the implementation of IPM is crucial to the success of the introduction of this crop in a new area. The information presented in the current review compiles experiences and strategies to develop an IPM, providing a valuable reference for any region worldwide [38,39,40,41,43,44,46,50].

The most important pests found in quinoa crops belong to the orders Hemiptera, Diptera, and Lepidoptera. Two main factors that complicate their control are their cryptic life styles (e.g., *A. karli* remains concealed in the stems *E. melanocampta* hides among the spun-together apical leaves) and their infestation of the crop during the grain filling stage, causing direct damage to the production (e.g., *Eurysacca* spp., *S. atripicella*, *Nysius* spp., and *L. simulans* feed on the grains). Their presence and impact on quinoa are related to the prevailing environmental conditions, particularly the climatic and crop management conditions [34,44,82], and, therefore, different levels of incidence and damage are expected across the season in the countries and regions where these pests occur.

Some major pests reported in the Andean region, such as *L. huidobrensis, L. hyalinus*, *M. euphorbiae, M. persicae*, and *F. occidentalis* are of cosmopolitan distribution, and others such as *S. eridania*, *S. recurvalis*, and *C. virescens*, are present in different parts of the world [24,33]. Therefore, given the circumstances, these species may eventually emerge as pests in quinoa crops cultivated around the world. On the other hand, there are species in other countries with similar behavior to those reported in the Andean region, such as *S. atripicella*, that cause similar damage to *E. melanocampta* or *N. simulans*, the damage of which is similar to that caused by other species of *Nysius* elsewhere [34,40,50]. Experience with the management of South American pest species may help quinoa farmers from other areas augment their strategies for pest suppression.

The main obstacle to implementing effective IPM programs for quinoa around the world is the knowledge gap concerning the harmful and beneficial entomofauna associated with this crop. Improving insight into the impact of both groups would allow us to optimize the management of arthropod pests, from their sampling to the application of adequate control measures. More in particular, setting up economic threshold levels (ETLs) is an aspect that needs to be addressed, as these are essential components of an IPM [52]. Even in the Andean region, where quinoa has been cultivated for centuries, there are few references on ETLs for the major pests of the crop [32]. In this context, studies on the seasonal phenology of the key quinoa pests can generate useful basic information to better understand the status of the pests and to set up or fine-tune ETLs.

In areas where quinoa is recently introduced, timely on-plant sampling in combination with the use of traps is vital to detect initial infestations and first generations of a pest, and also to perform consecutive monitoring throughout the crop phenology. Even in regions where there have been no reports of a certain pest, it is worth setting up an effective sampling plan to monitor the occurrence of phytophagous arthropods on the different plant parts [14,52]. This is illustrated by the experience of North American farmers who failed to notice the presence of *A. karli* until this pest became a serious problem because the larvae remain hidden inside the stem [44].

In South America, quinoa is visited by a diversity of natural enemies (parasitoids and predators), which are part of the arthropod community in the agroecosystem [67]. This provides good perspectives for establishing CBC strategies that may improve quinoa production and may be particularly valuable in organic quinoa [73,74,83,84,85]. For conventional quinoa, the protection of functional biodiversity should focus on reducing the impacts of agrochemicals by promoting the use of environmentally friendly pesticides in the framework of an IPM [26,32,61,68]. Another important aspect of CBC to consider is the conservation of natural and semi-natural habitats around quinoa fields, which may serve as a refuge for natural enemies [86]. Outside the Andean region, studies on the diversity of natural enemies are essential to identify the key predators and parasitoids that may play a role in biological control strategies, including augmentative releases.

For chemical control, pyrethroids are widely used in areas where quinoa is cultivated [34]. However, this broad-spectrum insecticide has a short- and long-term negative effect on natural enemies in quinoa fields [61], making these and other broad-spectrum insecticides unsuitable for an IPM program. This situation is worsened in some areas of South America where such compounds are being applied without a rotation system with active ingredients having a different mode of action, which may lead to problems of insecticide resistance [65,87,88,89]. The use of selective insecticides should be prioritized, taking into account the composition of the local insect communities and considering prevailing pesticide regulations [90,91,92,93].

The implementation of IPM strategies in quinoa may differ according to the experience with the cultivation of the crop. Outside of the Andes, some regions have only recently introduced quinoa, while in others, this grain has been grown for production for more than twenty years; in the latter areas, varieties adapted to local conditions have already been studied and more adequate cultivation techniques have been developed. Nonetheless, there are challenges to face in all regions around the world where quinoa cultivation is in continuous expansion.

Agricultural extensionists play a crucial role in establishing IPM programs for quinoa, conveying research data on pests to quinoa farmers, and assisting them in managing emerging pest issues. Institutes related to agrarian regulations for each country have an important role to play by establishing appropriate agricultural policies, particularly in areas where there is a lack of technical regulations for key cultivation parameters (e.g., sowing periods per locality, pesticides allowed, and appropriate varieties per zone).

Due to the increasing popularity of quinoa, studies on the arthropofauna associated with the crop have been increasing over the last two decades. However, there is a knowledge gap for the major pests that needs to be addressed, particularly in areas at lower altitudes than the traditional zones in the Andes, where the pest complex is more extensive. This may help to fine-tune the strategies of pest management. Additionally, given that quinoa growers from the Andes cultivate this crop in plots of less than 2 ha [15,16,18], the organization of farmers is crucial to apply strategies of IPM at a larger scale in order to have a major impact on pest prevalence and crop losses. In new production zones outside of the Andean origin, the arthropofauna is not completely understood, therefore studies on arthropods associated with quinoa in these areas will set the basis for successful IPM programs [28,55]. Cultivating quinoa in new geographical contexts may influence the diversity and abundance of local arthropods which may also affect other crops; therefore, it is relevant to understand the dynamic relationships between the selected quinoa varieties and emerging pest complexes. Future research could prioritize elucidating these interactions between quinoa varieties and evolving pest communities in diverse geographical contexts.

One of the primary challenges in the successful global production of quinoa is the development of varieties that fit the local characteristics. This includes cultivars adapted to specific environmental conditions and resistant or tolerant to certain pests and diseases. Moreover, given its remarkable resilience and adaptability, quinoa stands out as a promising candidate for ensuring food security amid climate change. This potential is based on the various stress tolerance mechanisms identified in the crop. Presently, global efforts in quinoa breeding aim to enhance integrated pest management (IPM) programs, further supporting its cultivation and sustainability [94,95,96].

## Figures and Tables

**Figure 1 insects-15-00540-f001:**
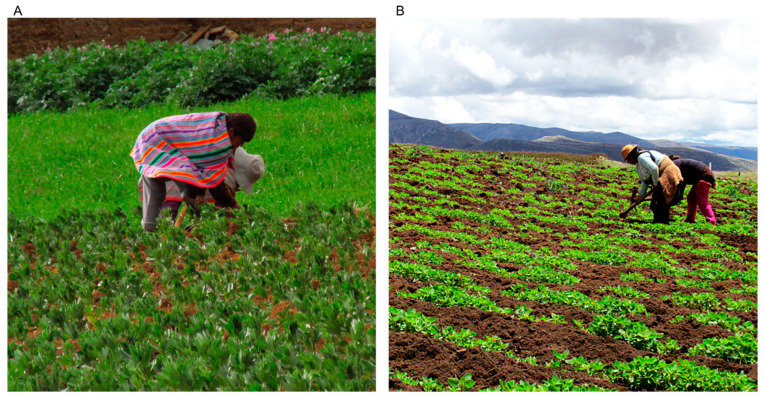
Farmers from the department of Huancavelica (in the highlands of Peru) cultivating quinoa under two cultivation systems: (**A**). crop association, with legumes; (**B**). monoculture (Photos: author).

**Figure 2 insects-15-00540-f002:**
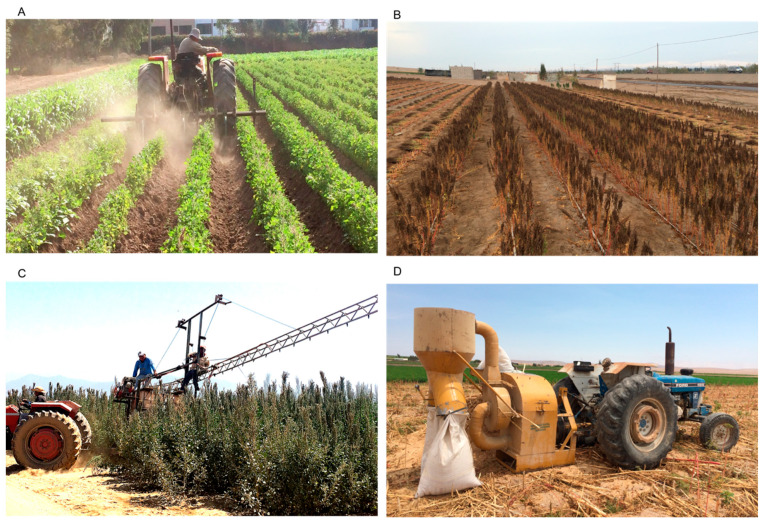
Quinoa cultivation in non-traditional systems: (**A**). cultivation and hilling; (**B**). drip irrigation; (**C**). chemical control with spray equipment; and (**D**). threshing (Photos: author).

## Data Availability

This article has no additional data.

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
