# Peer review of "Advances in the Integrated Pest Management of Quinoa (Chenopodium quinoa Willd.): A Global Perspective"

_insects, 2024, doi:10.3390/insects15070540_

Round 1

Reviewer 1 Report

Comments and Suggestions for Authors

The paper reviews quinoa pests in the world and how to manage these pests with an integrated control management strategy.  Pests and natural enemies are listed, and different control methods are discussed.

 Please include the family to which quinoa belongs.

The eight principles of IPM are given but also include a definition for IPM.

Pheromone traps Line 277 to 282.

The use of 4 traps per hectare is recommended for three species. Please clarify. Are four traps used for each species?

Reviewer 2 Report

Comments and Suggestions for Authors

In general, this review manuscript is well-written and supported by a total of 94 related articles. Explanation of the article Quinoa is a grain originally from the Andes that has become popular worldwide due to its nutritional benefits. In the meantime, the cultivation of quinoa has spread beyond its native Andean region and is now growing worldwide. At the same time, new pests have begun to attack the quinoa crops, leading to significant problems and production losses. Based on data (Table S1), the pests presented in this article belong to the orders Orthoptera, Hemiptera, Thysanoptera, Diptera, Coleoptera, Lepidoptera, Hymenoptera, etc. The report deals with the status of these pests worldwide and the agricultural strategies to control them within the framework of integrated pest management (IPM). The report also includes a report on global pest infestations and considerations for implementing IPM for quinoa. However, when reading the article, I have the impression that it primarily only reports on global problems related to pest infestations and the techniques available to control these pests. A more critical argument/discussion in the Discussion Section should be presented more thoroughly. The following aspects are suggested to improve the article:

(i) Pests:

Based on the data, what types of pests should be emphasised? For example: Hemiptera? Orthoptera? If Hemiptera or Orthoptera, why can these insects cause severe damage to quinoa?

Why have these pests become a new threat to quinoa? Is it related to their morphological characteristics that they become an important pest in the respective areas/regions/countries?

Are their presence and impact related to factors such as climate, soil conditions, etc.? Are there different types of pests and infestations in different countries and regions?

(ii) Integrated Pest Management (IPM):

What are the main obstacles to implementing effective IPM programmes for quinoa around the world? Are the obstacles the same?

IPM should consider not only the eight principles but also the impact on the environment and the reasons for different success rates in different areas. IPM strategies need to be tailored to specific locations and pests, taking into account the local environment and available techniques. Please critically discuss the IPM aspects.
